# New Coumarin Dipicolinate Europium Complexes with a Rich Chemical Speciation and Tunable Luminescence

**DOI:** 10.3390/molecules26051265

**Published:** 2021-02-26

**Authors:** Sebastiano Di Pietro, Dalila Iacopini, Aldo Moscardini, Ranieri Bizzarri, Mauro Pineschi, Valeria Di Bussolo, Giovanni Signore

**Affiliations:** 1Department of Pharmacy, University of Pisa, Via Bonanno Pisano 6, 56126 Pisa, Italy; mauro.pineschi@unipi.it (M.P.); valeria.dibussolo@unipi.it (V.D.B.); 2Department of Chemistry and Industrial, University of Pisa, Via G. Moruzzi 3, 56126 Pisa, Italy; dalila.iacopini@phd.unipi.it; 3NEST, Scuola Normale Superiore and Istituto Nanoscienze-CNR, Piazza San Silvestro 12, 56127 Pisa, Italy; aldo.moscardini@sns.it (A.M.); ranieri.bizzarri@unipi.it (R.B.); 4Department of Surgical, Medical and Molecular Pathology, and Critical Care Medicine, University of Pisa, Via Roma 55, 56126 Pisa, Italy; 5Fondazione Pisana per la Scienza, Via F. Giovannini 13, 56017 San Giuliano Terme (PI), Italy

**Keywords:** europium complex, coumarin, dipicolinate, chemical speciation, luminescence

## Abstract

Europium (III) luminescent chelates possess intrinsic photophysical properties that are extremely useful in a wide range of applications. The lack of examples of coumarin-based lanthanide complexes is mainly due to poor photo-sensitization attempts. However, with the appeal of using such a versatile scaffold as antenna, especially in the development of responsive molecular probes, it is worth the effort to research new structural motifs. In this work, we present a series of two new tris coumarin-dipicolinate europium (III) complexes, specifically tailored to be either a mono or a dual emitter, tuning their properties with a simple chemical modification. We also encountered a rich chemical speciation in solution, studied in detail by means of paramagnetic NMR and emission spectroscopy.

## 1. Introduction

In the last decades, luminescent lanthanide complexes, in particular of europium (III), have attracted great interest for a large range of applications [1,2], which take advantage of their peculiar emission properties (large Stokes shift, long excited-state lifetime, specific spectral fingerprint) to solve a wide scope of physicochemical and biological problems [3,4]. The possibility to temporally decouple probe emission from the autofluorescent background of biological specimens led to important synthetic efforts in recent years to create responsive lanthanide probes for bioimaging applications. In this context ratiometric probes are particularly attractive due to their straightforward implementation in analytical methods and their easy-to-read output [5]. Lanthanide-based ratiometric probes can be designed exploiting the photophysical properties of the organic chromophore (antenna) and lanthanide ion. Notably, efficient interaction relies on an extremely delicate balance of factors (energy matching, antenna triplet state production, shielding from solvent O-H oscillators, photo and chemical stability) which are mainly antenna/ligand dependent [6].

Coumarin-based probes are known to be incredibly versatile systems with photophysical properties dynamically modulated by a large set of boundary conditions (pH, polarity, viscosity, specific analyte) [7,8,9,10]. However, their use as a lanthanide sensitizer has been limited to just a few examples [11,12,13,14,15,16,17,18,19,20,21,22,23,24]: of note, those cases where the coumarin antenna directly coordinates the lanthanide ion are usually characterized by poorly stable complexes with limited application [11,14,21], whereas in all the other examples the absence of a tight connection with the metal ion greatly hampers the efficiency of the sensitization process (i.e., low quantum yields) [12,15,16,18,19,20,22,23,24]. Moreover, coumarin-based lanthanide complexes, due to the antenna intrinsic photophysical properties, usually display contemporary emission of both the lanthanide ion and the coumarin fragment: although the dual emission is an intriguing feature considering possible applications, it is fundamental to conceive a system which can be easily modified to finely tune the emission properties. With all this in mind, we designed a new series of coumarin-dipicolinate ligands (Scheme 1) which merges the photophysical properties of the coumarin chromophore with the known stability of the tris dipicolinate (dpa) lanthanide complexes in a fully conjugated structure. In more detail, we here describe the synthesis of the two novel ligands (**HL_1_** and **HL_2_**) and their relative tris europium complexes (**Na_3_[Eu(L_1_)_3_]** and **Na_3_[Eu(L_2_)_3_]**) with a straightforward and versatile synthetic pathway. These two new ligands have been specifically tailored to exhibit single or dual emission properties using one unique structural motif: passing from **HL_1_** to **HL_2_**, the insertion of an iodine atom on the coumarin core (in C-6 position), efficiently promotes the triplet state production, suppressing the dual emission without altering the complex scaffold.

## 2. Results

### 2.1. Ligands and Tris Complexes Synthesis

The synthetic pathway for the preparation of the new ligands **HL_1_** and **HL_2_** and the relative tris europium complexes **Na_3_[Eu(L_1_)_3_]** and **Na_3_[Eu(L_2_)_3_]** was designed around the key intermediate triester **3** that made it possible to conjugate virtually any coumarin scaffold to the dpa ring with simple and inexpensive chemistry. This planning enabled a highly versatile synthesis, because the only requirement was that the aromatic/heteroaromatic portion needed for the coumarin construction possessed a salicyl aldehyde-like functionality. As aforementioned in the introduction, in this work we focused on a simple 2*H*-chromen-2-one core and its 6-iodo derivative, and their synthesis has been realized as follows.

The pathway started from commercially available chelidamic acid (Scheme 1), which was converted to the 4-chloro diester compound **1** through the DMF activated reaction of the pyridine ring with thionyl chloride. This approach provided the concomitant formation of the two diester functionalities. Compound **1**, an electron poor heteroaromatic ring, was then transformed into the tetraester derivative **2** by means of an aromatic nucleophilic substitution with diethyl malonate in DMSO using cesium carbonate as base. Thanks to the properties of α-dicarboxylic acids, compound **2** was converted into the trimethylester **3**, with an initial cleavage of the four ester functionalities with NaOH in methanol, decarboxylation through acidification and final re-esterification, all in a single step.

The coumarin core of compounds **4** (6-H) and **5** (6-iodo) was then assembled taking advantage of the reported smooth reactivity of hydroxybenzaldehydes-like scaffolds with β-aryl esters in Knoevenagel conditions [8,9,10]: condensation with salicylaldehyde or with iodo-salicylaldehyde led to the formation of intermediates **4** and **5** in excellent yields. A simple ester hydrolysis of **4** and **5**, followed by precipitation from acidic water, concluded the synthetic pathway: target ligands **HL_1_** and **HL_2_** were accordingly obtained in 31.5% and 22% overall yield over 5 steps, respectively.

The self-assembled tris europium complexes of **HL_1_** and **HL_2_** were prepared in water by controlled deprotonation of the two dicarboxylic acids with NaOH (keeping the pH below 8), followed by the addition of 0.33 eq of europium chloride hexahydrate. The solution was filtered through cellulose acetate filters (to get rid of traces of insoluble 1:2 complex, see next section for details) and then concentrated. The tris complexes **Na_3_[Eu(L_1_)_3_]** and **Na_3_[Eu(L_2_)_3_]** precipitated from the concentrated solution as trisodium salts.

All the intermediate compounds were characterized by ^1^H- and ^13^C-NMR and the two ligands **HL_1_** and **HL_2_**, and the two tris complexes **Na_3_[Eu(L_1_)_3_]** and **Na_3_[Eu(L_2_)_3_]** were fully characterized by ^1^H- and ^13^C-NMR, high resolution mass spectrometry (HRMS) and their purities assessed by HPLC at 254 nm (purity > 95% for all the target compounds) and by elemental analysis for the two target complexes.

Next, we sought to evaluate the chemistry of **L_1_^2−^** and **L_2_^2−^** anions with Eu^3+^ in terms of chemical speciation taking advantage of both paramagnetic NMR and emission spectroscopy.

### 2.2. Chemical Speciation of the Eu/L_1/2_ System

#### 2.2.1. Paramagnetic NMR

The synthesis, isolation and characterization of both the tris europium **L_1_^2−^** and **L_2_^2−^** homoleptic complexes were clear and straightforward as reported in the previous section. Interestingly, for different stoichiometries with respect to Eu:Ligand = 1:3, we encountered a rich solution chemical speciation, which has been studied in detail by means of paramagnetic NMR, emission spectroscopy and HRMS, revealing an uncommon chemical behavior for DPA-like complexes [25,26,27,28,29,30].

Solution NMR is a powerful tool for the structural analysis of paramagnetic lanthanide complexes [31,32,33,34,35,36,37,38]: the paramagnetism of a lanthanide trivalent ion influences both the chemical shift and linewidth of the resonance of a nucleus close in space. In particular, the paramagnetic contribution (in sign and magnitude) to the chemical shift has dipolar origin, and it correlates with the relative position (in polar coordinates) of the nucleus with respect to the lanthanide ion and the major symmetry axis of the system, through the McConnel-Robertson equation [39] (Equation (1), valid for *C*_3_ systems or higher).
(1)$$\delta_{\mathrm{para}}=D_{Ln}\frac{3\cos^{2}\theta -1}{r^{3}}$$

In this equation, *θ* and *r* are the polar coordinates of the nucleus with respect to the lanthanide ion in the origin and the major symmetry axis, and *D_Ln_* is a lanthanide dependent constant (which arises from the Bleaney’s theory [40]) that determines the spreading of the ligand resonances upon lanthanide influence, both in magnitude and in sign, starting from the diamagnetic free ligand values (or La/Lu complex). This apparently simple equation is a source of structural information: for europium *D_Eu_* has a positive value and a magnitude relatively small, especially with respect to other paramagnetic lanthanides such as Dy, Yb, Tm or even Tb. In addition, the paramagnetic contribution (which is of course distance-dependent) vanishes if the nucleus is located at a polar angle of 54.7°, and in particular, for europium, will be negative for 54.7° < *θ* < 109.4° (with a maximum at 90°).

The comparison between the proton NMR spectrum in D_2_O of **L_1_** (triethylammonium salt) and **Na_3_[Eu(L_1_)_3_]** complex (Figure 1) can be rationalized, then, by paramagnetic analysis.

The proton spectrum of the isolated **L_1_** complex in deuterated water showed a unique set of six resonances as expected for a tris complex of *C*_3_ symmetry: all the proton observed shifts (δ_obs_) in the europium complex were upfield (shifted to lower δ) with respect to the free **L_1_^2−^** ligand. This reflects the structure and the symmetry of the tris system which is characterized by three ligand units wrapped around the Eu^3+^ ion, usually as a more or less distorted tricapped trigonal prism, collocating virtually all the ligand atoms in the part of space with 54.7° < *θ* < 109.4° polar angles. In addition, it was not surprising to found that the most paramagnetically shifted proton atoms were the two singlets H-Py (δ_obs_^Lig^ = 8.20 ppm, δ_obs_^Eu^ = 4.33 ppm) and H-4 (δ_obs_^Lig^ = 8.27 ppm, δ_obs_^Eu^ = 6.95 ppm), which are the closest in space to Eu^3+^, followed by all the others with less accentuated shifts. The resonance of H-Py has a diagnostic value: its paramagnetic shift of −3.87 ppm is perfectly in line with the value −4.23 ppm (δ_obs_ = 3.77 ppm) reported in D_2_O for Na[Eu(dpa)_3_] complex [29] (which is also in line with the reported XRD structure [41]), confirming the isostructurality between the two complexes for the dipicolinate fragment, both in terms of distances and skew angle of the pyridine ring with respect to the *C*_3_ axis (connected to *θ* of H-Py). Importantly, all the proton resonances of the complex were experiencing a moderate line broadening which is mainly due to the poor solubility of the **Na_3_[Eu(L_1_)_3_]** in water in the millimolar range of concentration requested for recording the NMR spectrum, rather than to the fast to moderate exchange chemical dynamics or paramagnetism effect (on *T*_2_).

A first attempt of NMR speciation study was performed by a titration (Appendix A) of the **L_1_^2−^** ligand (triethylammonium salt) in D_2_O with Eu^3+^ (chloride hexahydrate salt) which provided some interesting insights, but raised, at the same time, some questions. The 1:3 complex observation was again straightforward (around 0.3 eq of europium), even if the species was poorly soluble in water media as aforementioned, and the NMR spectrum displayed a moderate line broadening. At lower molar ratios (0.15 eq of europium) the spectrum was dominated by an intermediate/slow exchange regime in the NMR timescale, with large line broadening of both free and bound ligand resonances (one resembling the 1:3 species) involved in a chemical exchange. Most importantly, in deuterated water each addition of europium salt above the 1:3 ratio determined the precipitation of a yellow solid, that precluded any further rationalization at higher molar ratios.

More informative results were obtained by titration in deuterated DMSO, a solvent where all the Eu^3+^/L_1_ species are perfectly soluble (Figure 2 for the 5–9 ppm window, Appendix A for the 1–4 ppm one).

Interestingly, the NMR study in *d*_6_-DMSO revealed a rich chemical dynamic which involved at least four species: free **L_1_^2−^**, the 1:3 complex and two additional coordination compounds for number of equivalents below and above the tris system stoichiometry. All these species were in a slow exchange regime in the NMR timescale (in contrast with Ln/DPA [29]), with only small/moderate line broadening, mainly due to ongoing chemical dynamics phenomena. Overall, this analysis availed the following key observations and remarks:

At 0.35 eq there was the unique presence, as expected, of the trianionic 1:3 complex **[Eu(L_1_)_3_]^3−^** (the spectrum was almost identical to that recorded in D_2_O). The spectrum appeared relatively sharper with respect to the one in water media due to its greater solubility in DMSO.

Titration points above 0.35 eq showed the slow/intermediate exchange, between 0.4 and 0.65 eq, among the 1:3 species and a new one, which remained unique and almost unaltered in resonances shifts even until 4 eq addition of Eu^3+^ (the last points showed a small broadening due to the presence in solution of amounts of free paramagnetic Eu^3+^). This new species has been assigned to the homoleptic monoanionic 1:2 complex **[Eu(L_1_)_2_]^−^**: its H-Py resonance was at δ_obs_ = 6.16 ppm with a −2.23 ppm paramagnetic shift. The solid precipitating from water at similar stoichiometry ratio could be then referred to the 1:2 complex, which is less soluble compared with the tris compound, considering that it is a mono-charged system.

Interestingly, in the 0.1–0.3 eq interval we came across the presence of another Eu/L_1_ species; the 0.2 eq spectrum was, at first glance, a complex characterized by a set of four resonances (together with a small amount of residual **L_1_^2−^** ligand): H-Py with δ_obs_ = 5.00 ppm (δ_para_ = −3.2 ppm), H-4 with δ_obs_ = 7.26 ppm (δ_para_ = −1.01 ppm), a doublet (δ_obs_ = 7.38 ppm) and a triplet (δ_obs_ = 7.18 ppm). In addition, these last two signals integrated both for 2 protons but, while the doublet was quite sharp, magnifying the triplet peak was possible to appreciate the superimposition of the H-6 and H-7 signals (then the benzene ring is only an apparent AA’BB’ system, but a ABCD one as expected). The overall paramagnetic shift for all the nuclei, with respect to the 1:3 complex, was still upfield, but less pronounced in magnitude. The 0.3 eq point showed the presence of both this new species and the 1:3 complex in slow exchange. On account of these findings (this species clearly existed between 0.2 and 0.3 eq of Eu^3+^), we considered the possibility of a first example of tetrakis dpa-like europium complex, the pentanionic **[Eu(L_1_)_4_]^5−^** (formula indicative of the sole stoichiometry).

The choice of using an organic counterion (triethylammonium) in the NMR titration allowed us to further comprehend the nature of these species. In axial complexes (*C*_3_ or higher) it is common to expect the dynamic binding of counterion, water, analyte molecule/s through an exchangeable axial coordination site [32]. The 1–4 ppm window of the titration spectra is reported in Appendix A: the signals of the ethyl chain for Et_4_NH^+^ in **L_1_^2−^** were located respectively at δ = 2.87 ppm (CH_2_) and δ = 1.09 ppm (CH_3_); in 0.1 and 0.2 eq spectra both these resonances were gradually shifted downfield (CH_2_ by 0.3 and >1 ppm, CH_3_ by 0.15 and 0.3 ppm), to suddenly go back towards more moderate positive shifts (CH_2_ by 0.2 ppm and CH_3_ by 0.1 ppm) between 0.3 and 0.4 eq. Above 0.4/0.5 eq the positive shift slightly decreased but remained similar to 0.3/0.4 eq interval spectra. In addition, between 0.3 and 1 eq the CH_2_ signal was a superimposition of two quartets (while the CH_3_ is always a sharp triplet), while above 1 eq the CH_2_ signal shape was again a unique quartet. This behavior is in line with the nature of europium species found in the titration. The counterion is in fast exchange between free and bonded form, and the resonances shifts are the weighted average of the two possibilities. In addition, the europium ion in axial complexes (*C*_3_ or higher) shifts downfield molecules bonded along the symmetry axis. In the 0.2/0.5 eq interval the peaks strongly shifted downfield because the tetrakis **[Eu(L_1_)_4_]^5−^** is a highly charged pentanionic complex and the bonded form of Et_4_NH^+^ is predominant and the weighted average of the two is shifted towards the bonded one. However, in the 0.3/1 eq interval both the tris **[Eu(L_1_)_3_]^3−^** and the bis **[Eu(L_1_)_2_]^−^** complexes are gradually less charged and the weighted average chemical shift of the two forms of Et_4_NH^+^ moves still in positive, but with a smaller magnitude because the bonded form is not predominant anymore. The presence of two CH_2_ quartets between 0.3 and 1 eq reflects the contemporary presence of tris and bis species as previously observed (Figure 2).

While the rationalization of the paramagnetic shifts for the bis **[Eu(L_1_)_2_]^−^** complex can be tricky, due to the lower symmetry (for *C*_2_ the simplified version of Equation (1) is no longer valid and a rhombic term has to be considered), for the tetrakis **[Eu(L_1_)_4_]^5−^** complex additional details could be elaborated: the overall less pronounced negative shifts for all the proton resonances reflects the steric requirements of a tetrakis species with fourfold axial symmetry (most likely a distorted square antiprism polyhedron), which is characterized with ligand units arranged in space, with skew angles, with respect to symmetry axis, of the coplanar pyridine/coumarin conjugated rings smaller than the one of the 1:3 complex. This geometric constraint is mainly translated to polar angles *θ* for each nucleus much closer to 54.7° (or 125.3°) value (of zero δ_para_) and then into smaller negative paramagnetic shifts in magnitude.

#### 2.2.2. Emission Spectroscopy and HMRS

Lanthanide centered emission spectroscopy, in particular for europium complexes, is a large source of information for the intimate connection between speciation and symmetry of the system and luminescence profile of the ^5^D_0_ → ^7^F_J_ transition bands [42,43].

In Appendix A are reported the fluorescence titration of **L_1_^2−^** (50 µM solution in Tris HCl buffer at pH = 7.4, exciting at 360 nm) with Eu^3+^ (chloride hexahydrate salt solution in water) and the relative titration curve, monitoring the fluorescence output at 615 nm on the ^5^D_0_ → ^7^F_2_ hypersensitive emission band. The variation in intensity of the 615 nm band upon the addition of europium, is a clear indication of the formation of the 1:3 complex **[Eu(L_1_)_3_]^3−^**. Indeed, fluorescence reached its maximum at 0.3 eq of Eu^3+^, followed by a constant decrease for the formation of the 1:2 complex **[Eu(L_1_)_2_]^−^** which has a lower propensity to luminesce: in particular, in the 0.45–0.6 eq interval, where tris and bis species coexist, the decrease was initially less steep (0.6 eq point was somewhat out of the trend, but it falls within the experimental error of measurement). Moreover, the ligand residual fluorescence (centered at 420 nm), present in the case of **L_1_**, steadily decreased upon europium addition. This is in line not only with the formation of the europium complexes, but also, after 0.5/0.6 eq, with the quenching effect due to increasing amounts of free europium in solution.

In the same conditions (solvent, concentration), we monitored the luminescence decays (function of the lifetime τ) on the 615 nm band (λ_ext_ = 360 nm) for **L_1_^2−^** upon addition of Eu^3+^. Notably, this experiment allowed us to separate the contribution of multiple europium species to the luminescence, in agreement with a multiexponential decay fit (Equation (2)).
(2)$$I\left(t\right)=\overset{n}{\underset{1}{\sum }}I_{0,n}e^{-\frac{t}{\tau_{n}}}$$

The total intensity at a certain instant of the experiment is the sum of the contributions of the *n* emitting species, each one characterized by its intensity at zero time *I*_0_ and its lifetime *τ_n_*. In Appendix A are reported the fitted lifetime curves for each titration point and from Appendix A all the individual fitting (adjusted R^2^ ≥ 0.9999). This analysis gave us further insights on the speciation.

In the 0.05/0.25 eq interval we observed the monoexponential decay of tetrakis species characterized by a lifetime of 0.73 ± 0.01 ms, while between 0.35 and 0.5 eq the contemporary presence of tris and bis complexes with lifetimes of 0.6 ± 0.1 ms and 0.24 ± 0.1 ms, respectively. The initial intensities *I*_0_ obtained from each fitting (mono- or biexponential) correspond to the single contribution of the species to the ^5^D_0_ → ^7^F_2_ 615 nm band. Normalized intensities *I*_0_, are reported in Figure 3, and allowed us to depict a good picture of the speciation in water (Tris-HCl at pH = 7.4).

In Figure 4 are reported the fluorescence spectra (focused on the europium emission bands) in DMSO of Eu^3+^/**L_1_^2−^** mixture at 1:2, 1:3 and 1:4 molar ratios (triethylammonium salt).

At 1:2 molar ratio, the predominance of **[Eu(L_1_)_2_]^−^** was clearly confirmed by the aspect of both ^5^D_0_ → ^7^F_1_ and ^5^D_0_ → ^7^F_2_ bands with a higher multiplicity (with respect to the other spectra) in accordance with the expected degeneracy of these transitions in *C*_2_ symmetry [42]. For 1:3 and 1:4 molar ratios, the spectra were almost superimposable and both resembled the [Eu(DPA)_3_]^3−^ one [30,43]: however, the intensity of these two spectra was almost identical. Considering the lower content of europium in the 1:4 solution than in the 1:3 one (and the **L_1_^2−^** concentration was the same) this could signal the presence of a slightly more emitting species (the tetrakis), at least in DMSO solution. In addition, we performed the same experiment with 1:4, 1:3 and 1:2 solution of Eu^3+^:**L_2_^2−^** in DMSO (Appendix A), obtaining a similar fluorescent profile for all of them: the different bands deeply recalled the tris complex shape (very similar to [Eu(DPA)_3_]^3−^ as aforementioned), but with different intensities (1:3 > 1:4 > 1:2), depicting a more simplified situation for **L_2_** with the predominance of the tris complex all over the different molar ratios.

Last, looking for a further validation of these observations, we analyzed these three different molar ratio mixtures in DMSO for **L_1_** and **L_2_**, with a high resolution mass spectrometer (HRMS): the 1:2 solution of both the ligands showed the presence of the [M − H]^−^ peaks of **[Eu(L_1_)_2_]^−^** at *m*/*z* = 770.97748 (calculated for C_32_H_14_EuN_2_O_12_
*m*/*z* = 770.97646, Appendix A) and the [M − H]^−^ peaks of **[Eu(L_2_)_2_]^−^** at *m*/*z* = 1022.77087 (calculated for C_32_H_12_EuI_2_N_2_O_12_
*m*/*z* = 1022.76975, Appendix A) respectively, along with the tris complex presence. Unfortunately, for the 1:4 solution (only **L_1_** reported in Appendix A) the only peaks found have been assigned to bis and tris europium complexes (and free ligand), with no trace of signals connected to a tetrakis species, even in the mildest ionization conditions. This could be due to the labile nature of this species in the HMRS experimental condition.

We are currently working on the isolation and characterization of this species of 1:4 stoichiometry, which at the time of this work was still not considered perfectly clear: a europium complex of coordination number of 12 is quite unusual (CN = 10 is more common for Eu^3+^), but it could be possible in space as a tris complex axially capped with an additional ligand unit (most of the findings can also be explained in this way). Last, we could also be dealing with a different hydration state of the tris complex in the 0.05–0.33 eq range, but this assumption is less corroborated by the photophysics of the tris complexes and the one typical of europium tris dipicolinate. All these hypothesis are currently under evaluation.

### 2.3. Photophysics of Na_3_[Eu(L_1_)_3_] and Na_3_[Eu(L_2_)_3_] Complexes

The absorption and, most importantly, the emission spectroscopic properties of the isolated and characterized tris complexes **Na_3_[Eu(L_1_)_3_]** and **Na_3_[Eu(L_2_)_3_]** are reported and elucidated in the following.

In Figure 5 are depicted the absorption spectra in water of **Na_3_[Eu(L_1_)_3_]** and **Na_3_[Eu(L_2_)_3_]** complexes and the relative ligand anions **L_1_^2−^** and **L_2_^2−^** (triethylammonium salts): both ligand anions and tris complexes showed an absorption profile between 220 and 400 nm with three maxima due to different ligand centered π → π* transitions. Some differences, as expected, were appreciable between the two different ligands/complexes, while the absorption spectra of free and complex organic molecules were almost superimposable in terms of frequency (small red shift of 5–10 nm from ligands to complexes), except for the threefold increase in intensity due to the complex formation. **Na_3_[Eu(L_1_)_3_]** and **L_1_^2−^** were characterized by a 237 nm (ε = 97,000 L mol^−1^ cm^−1^ for the complex, ε = 16,000 L mol^−1^ cm^−1^ for the free ligand), 308 nm (ε = 43,800 L mol^−1^ cm^−1^ for the complex, ε = 17,400 L mol^−1^ cm^−1^ for the free ligand) and 330 nm peak (ε = 41,100 L mol^−1^ cm^−1^ for the complex, ε = 15,000 L mol^−1^ cm^−1^ for the free ligand); **Na_3_[Eu(L_2_)_3_]** and **L_2_^2−^** both showed a 20 nm bathochromic shift (red shift) for the lowest energy peak, in particular were characterized by a 237 nm (ε = 123,500 L mol^−1^ cm^−1^ for the complex, ε = 23,600 L mol^−1^ cm^−1^ for the free ligand), 302 nm (ε = 43,000 L mol^−1^ cm^−1^ for the complex, ε = 15,600 L mol^−1^ cm^−1^ for the free ligand) and 350 nm peak (ε = 21,500 L mol^−1^ cm^−1^ for the complex, ε = 7400 L mol^−1^ cm^−1^ for the free ligand). An interesting comparison could be done with data reported by Chauvin et al. in 2013 [20] for a series of tris 4-substituted dpa-based europium complexes with PEG spacers bearing coumarins scaffolds. In that work, dpa and coumarin rings were isolated chromophores (separated by 3 PEG units connected to C4 or C7 of the coumarin ring), and the relative complexes showed a 220–350 nm absorption span (with two maxima in the 300–350 nm zone similar to our complexes), which is smaller with respect to our 220–400 nm range. In addition, our extinction values for the complexes are slightly bigger than the ones reported in Chauvin’s paper (20,000–40,000 in the 300–400 nm zone for our complexes vs. 20,000–30,000 in the 300–350 nm range for their system). These peculiar differences are dictated by the conjugation between coumarin and dpa rings, with respect to those isolated by Chauvin et al.

In Figure 6 are depicted the luminescence spectra in water of **Na_3_[Eu(L_1_)_3_]** and **Na_3_[Eu(L_2_)_3_]** complexes excited at 360 nm, together with the relative excitation and absorption spectra. Note that the successful sensitization of the europium ion excited energy states through the excitation of the ligand, at a wavelength as 360 nm, is a confirmation of the presence of an extended π-electron system of the coumarin dipicolinate rings as a unique antenna/ligand entity. The metal centered emission for both complexes showed an almost superimposable profile for the ^5^D_0_ → ^7^F_J_ set of transitions, which is in line for shape and multiplicity with a *C*_3_ symmetry europium complex [42,43], and almost identical to the [Eu(DPA)_3_]^3−^ fluorescence spectrum: a small, but important, difference was the presence, for both **L_1_** and **L_2_** tris complexes, of the weak 580 nm emission band due to the ^5^D_0_ → ^7^F_0_ transition, which is, on the contrary, absent in *D*_3_ symmetry for the tris dipicolinate europium complex [30,43]. However, the most glaring difference in emission behaviors was, as planned, the successful suppression of the residual ligand-centered fluorescence for the **L_2_** tris complex with respect to the dual emission for the **L_1_** one: it is not unusual for coumarin antennas to display their residual ligand centered fluorescence together with the metal centered one [18,20,23,24]. Coumarin is a performant and dynamic emitter [7] which is capable of sensitizing europium (and other lanthanides) ions through an excited triplet state, mainly thanks to the strong effect of such a heavy ion in favoring the intersystem crossing process [44]. Noteworthy, it is crucial to underline that for coumarin-based antennas, the quality of the energy transfer process and the effect of the latter on the quantitative sensitization of europium is not only a matter of energy matching (which remains extremely important), but it is also deeply connected to the production of triplet from the excited singlet state, which for such an excellent fluorophore is not trivial. In addition, the residual ligand centered emission is far from an unwanted result: in fact, our goal has been to rationally design an antenna/ligand which, with a simple chemical modification (insertion of the iodine atom from **L_1_** to **L_2_**), could allow us to have access to both dual emission and europium centered systems. In addition, the dual emission of **Na_3_[Eu(L_1_)_3_]** can be easily modulated (time gating), considering the different time regime of these two processes (nanoseconds for ligand centered emission to hundreds of microseconds to a millisecond for the europium centered one): in Appendix A is reported the phosphorescence spectrum of **Na_3_[Eu(L_1_)_3_]** in water with a delay time of 200 µs which displayed only the europium emission bands. In fact, the europium radiative emission lifetimes (τ_obs_) for **Na_3_[Eu(L_1_)_3_]** and **Na_3_[Eu(L_2_)_3_]** isolated complexes in water were 0.61 ± 0.05 ms and 0.57 ± 0.05 ms, respectively: these values are smaller than the ones reported by Chauvin for the dipicolinate-PEG-coumarin system (1–1.6 ms) [20] which were instead in line with the [Eu(DPA)_3_]^3−^ ones [45]. The fitted lifetime curves (Appendix A) were perfectly monoexponential (R^2^ ≥ 0.999), strongly supporting the hypothesis that a unique europium emitting species is present in the solution (together with a unique ^5^D_0_ → ^7^F_0_ band which is our case).

The most evident effect of the presence of the heavy iodine atom in C-4 on the coumarin ring of **L_2_** was to suppress the contextual ligand-centered fluorescence centered at 425 nm with respect to the **L_1_** complex. It is fundamental to understand how this marked difference in emission behavior is translated to the europium photoluminescence quantum yields of these two complexes (all curve fitting with R^2^ ≥ 0.95 for the complexes and 0.998 for the reference, Appendix A). Indeed, **Na_3_[Eu(L_1_)_3_]** displayed in water Φ^Eu^ = 4.9 × 10^−4^ ± 1 × 10^−4^ (1.2 × 10^−4^ ± 1 × 10^−4^ in DMSO) for the europium and Φ^L1^ = 4.8 × 10^−3^ ± 5 × 10^−4^ (1.2 × 10^−3^ ± 5 × 10^−4^ in DMSO) for the **L_1_** fluorescence; **Na_3_[Eu(L_2_)_3_]** displayed in water Φ^Eu^ = 4.3 × 10^−4^ ± 5 × 10^−4^ (6.2 × 10^−4^ ± 1 × 10^−4^ in DMSO) for the sole europium centered fluorescence. Interestingly, these value were smaller than Chauvin’s system (Φ^Eu^ from 0.016 to 0.018 depending on the spacer length and coumarin ring) [20] excited a 320 nm, but they are in line with other literature examples, where coumarin antennas usually fail to sensitize europium in solution with a quantum yield above 2% [19]. Notably, considering that the emission maximum of **L_1_** in **Na_3_[Eu(L_1_)_3_]** was 420 nm (23,800 cm^−1^), it is unlikely that the relative triplet state lies [44] in the 2500–3500 cm^−1^ optimum range above the Eu^3+ 5^D_0_–^7^F_0_ fundamental emitting transition at 12,300 cm^−1^ [46], affecting the quality of the energy transfer. A more quantitative measure of the efficiency of the energy transfer process could be obtained through the sensitization parameter η_sens_ (see Section 3.6 for its definition and calculation): in water **Na_3_[Eu(L_1_)_3_]** showed η_sens_ = 0.32% and **Na_3_[Eu(L_2_)_3_]** η_sens_ = 0.39%; these value are lower than the 3–5% values for the Chauvin systems [18,20] which underlines the non-performant energy transfer from ligand to metal. Most importantly, suppression of the ligand centered fluorescence passing from **L_1_** to **L_2_** complex did not translate, in either case, to an increased photoluminescence quantum yield (except for DMSO) and sensitization: this finding underlines again how an excellent energy transfer for europium complexes is always a delicate balance of energy matching, ligand centered triplet production, chemical/photochemical stability and water molecule shielding. Clearly, other non-radiative pathways must be present for the **L_2_** complex. These findings and their rationalization represent, however, an important starting point for future optimization of the performance of this series of complexes, especially considering their simple chemical synthesis and the versatility of their intermediates. Finally, we must stress that the insertion of an iodine atom on the ligand has a small, yet important, effect on the potential applications of the complex. Indeed, the presence of a shoulder in the absorption spectrum at higher wavelengths leads to a nearly two-fold enhancement of absorption at 370–375 nm, that translates to a much higher brightness at biocompatible wavelengths despite the negligible increase in luminescence quantum yield.

## 3. Materials and Methods

All solvents and chemicals were purchased from Sigma-Aldrich and used without further purification. Silica gel 60 F254 sheets (Merck, Darmstadt, Germany) were used in analytical thin-layer chromatography (TLC). LC chromatographic separation was performed on an RfCombiflash (Teledyne ISCO, Lincoln, NE, USA) preparative purification system. Evaporation was performed in vacuo (rotating evaporator). Sodium sulfate was used as drying agent. 1D and 2D-NMR were recorded with a Bruker Avance III 400 spectrometer (Bruker, Billerica, MA, USA), using the indicated deuterated solvents. Chemical shifts are given in parts per million (ppm) (δ relative to residual solvent peak for ^1^H and ^13^C). HPLC purity of final ligands and complexes were determined using a Waters Alliance 2695 equipped with a 2420 dual wavelength detector (Waters, Milford, MA, USA) using the following parameters: column Phenomenex Luna C8 150 mm × 3 mm × 5 μm (Phenomenex, Torrance, CA, USA), mobile phases water/TFA 100/0.01 *v*/*v* and Acetonitrile 100 *v*/*v*. Retention times (HPLC, t_R_) are given in minutes. Compound HPLC purity was evaluated at 254 nm. The ESI-MS spectrum was recorded by direct injection at 7 μL min^—1^ flow rate in an Orbitrap high-resolution mass spectrometer (Thermo, San Jose, CA, USA), equipped with H-ESI source. The working conditions were as follows: negative polarity, spray voltage −3.2 kV, capillary temperature 290 °C, S-lens RF level 50. The sheath and the auxiliary gases were set at 28 and 4 (arbitrary units), respectively. For acquisition and analysis, Xcalibur 4.2 software (Thermo) was used. For spectra acquisition a nominal resolution (at *m*/*z* 200) of 140,000 was used. Absorption and fluorescence spectra were recorded in cuvettes with 1 cm optical path (Hellma, Müllheim, Germany) at 23 °C on a Jasco V550 spectrophotometer (Jasco, Easton, MD, USA) and a Cary Eclipse spectrofluorometer (Varian, Palo Alto, CA, USA), respectively. The luminescence lifetime τ of the complex was measured by the phosphorescence plugin of the Cary-Eclipse spectrofluorometer. The luminescence decay was fitted to a mono/biexponential decay to recover τ_obs_. Elemental analyses have been performed on an Elementar (Elementar Analysensysteme GmbH, Langenselbold, Germany) Vario micro cube equipment.

### 3.1. Synthetic Procedures and Characterization

*2,6-Pyridinedicarboxylic acid, 4-chloro-2,6-dimethyl ester* (**1**). Chelidamic acid (15 g, 27.3 mmol, 1 eq) was dissolved in SOCl_2_ (120 mL) in a three-necked round-bottom flask and anhydrous DMF (1 mL) was added under N_2_ flow. The suspension was refluxed overnight. Upon completion the SOCl_2_ was removed by distillation under reduced pressure and the solid was suspended in MeOH (120 mL) at 0 °C. The resulting mixture was refluxed overnight. After cooling down, the product was crystallized and collected by filtration through a Buchner funnel, and washed with cold MeOH. Pure product **1** was obtained as a white solid (13 g, yield = 83%). ^1^H-NMR (400 MHz, CDCl_3_) δ (ppm) = 8.28 (s, 2H, Py), 4.02 (s, 6H, Me).

*2,6-Dimethyl 4-[2-ethoxy-1-(ethoxycarbonyl)-2-oxoethyl]-2,6-pyridinedicarboxylate* (**2**). 2,6-Pyridinedicarboxylic acid, 4-chloro-2,6-dimethyl ester **1** (5 g, 21.8 mmol, 1 eq) was dissolved in dry DMSO (100 mL) under N_2_ flow. Diethyl malonate (6.7 mL, 43.2 mmol, 2 eq) and Cs_2_CO_3_ (14.2 g, 43.6 mmol, 2 eq) were added and the resulting solution was stirred overnight at 100 °C. Then, the mixture was cooled down, diluted with water and acidified with HCl 1M until pH < 7. The aqueous solution was extracted with DCM and the organic phases washed with H_2_O and brine until neutral pH. After drying over anhydrous Na_2_SO_4_, filtration and evaporation of the solvent, product **2** was obtained as a solid (6.8 g, yield = 88%) which was used in the next step without further purification. ^1^H-NMR (400 MHz, CDCl_3_) δ (ppm) = 8.64 (s, 2H), 4.22–4.15 (q, ^3^*J* = 7.05 Hz, 4H), 4.01 (s, 1H), 4.00 (s, 6H), 1.29–1.24 (t, ^3^*J* = 7.03 Hz, 6H). ^13^C-NMR (150 MHz, CDCl_3_) δ (ppm) = 166.7, 163.9, 148.3, 137.15, 67.1, 61.5, 53.4, 14.0.

*2,6-Pyridinedicarboxylic acid, 4-(2-methoxy-2-oxoethyl)-2,6-dimethyl ester* (**3**). Compound **2** (6.8 g, 19.1 mmol, 1 eq) was suspended in 100 mL of MeOH, NaOH 1M (100 mL) was added and the resulting mixture was refluxed for 3 h. After cooling down and acidification with H_2_SO_4_ until pH = 1 (decarboxylation and re-esterification steps), the solution was stirred overnight at room temperature. Then, the solvent was evaporated and the solid was suspended in DCM and filtered through a Buchner funnel. The crude was purified by LC chromatography (80 g SiO_2_ column, solid state sampling, Hex/EtOAc gradient elution), affording pure product **3** as a white solid (3 g, yield = 60%). ^1^H-NMR (400 MHz, CDCl_3_) δ (ppm) = 8.21 (s, 2H, Py), 3.99 (s, 6H, Me), 3.77 (s, 2H, -CH_2_), 3.71 (s, 3H, Me). ^13^C-NMR (150 MHz, CDCl_3_) δ (ppm) = 169.5, 164.9, 148.5, 145.8, 128.9, 53.2, 52.6, 40.1.

General procedure for the synthesis of **4** and **5**. In a typical reaction, compound **3** (0.5 g, 1.9 mmol, 1 eq) was dissolved in 15 mL of DMSO dry under N_2_ flow. An appropriate amount of salicylaldehyde (170 μL, 1 eq, for 4) or iodo-salicylaldehyde (232 mg, 1 eq, for 5) and piperidine (1 eq) were added, and the mixture was stirred at 90 °C until complete conversion (HPLC). Solution was cooled down, the product precipitated and then it was centrifugated and washed with HCl 1M and finally dried under reduced pressure. Pure products were obtained in excellent yields.

Compound **4**. Yellow solid, yield = 80%. ^1^H-NMR (400 MHz, *d*_6_-DMSO) δ (ppm) = 8.72 (s, 1H, H-4), 8.66 (s, 2H, Py), 7.88–7.86 (d, ^3^*J* = 7.67 Hz, 1H, H-8), 7.73–7.69 (t, ^3^*J* = 7.34 Hz, 1H, H-7), 7.50–7.48 (d, ^3^*J* = 8.5 Hz, 1H, H-5), 7.46–7.41 (t, ^3^*J* = 7.66 Hz, 1H, H-6), 3.95 (s, 6H, -Me). ^13^C-NMR (150 MHz, *d*_6_-DMSO) δ (ppm) = 265.0, 159.5, 153.9, 148.4, 145.5, 144.3, 133.6, 129.9, 127.1, 125.4, 123.1, 119.5, 116.5, 53.3.

Compound **5**. Yellow solid, yield = 84%. ^1^H-NMR (400 MHz, *d*_6_-DMSO) δ (ppm) = 8.64 (s, 3H, H-4 + Py), 8.24–8.23 (d, ^3^*J* = 2.07 Hz, 1H, H-5), 8.00–7.96 (dd, ^3^*J* = 1.93 Hz, 8.56 Hz, 1H, H-7), 7.34–7.31 (d, ^3^*J* = 8.59 Hz, 1H, H-8), 3.95 (s, 6H, Me). ^13^C-NMR (150 MHz, *d*_6_-DMSO) δ (ppm) = 164.9, 159.0, 153.6, 148.5, 145.2, 142.8, 141.5, 137.8, 127.1, 123.9, 121.7, 118.9, 88.9, 53.3.

General procedure for the synthesis of **HL_1_** and **HL_2_**. In a typical reaction, diester **4** (100 mg, 0.29 mmol, 1 eq) or **5** (100 mg, 0.21 mmol, 1 eq) was dissolved in a 3:1 mixture of MeOH/H_2_O. Solid NaOH (15 eq) was added and the mixture was refluxed for 2 h. Upon completion (HPLC), MeOH was evaporated and the solution was acidified with HCl 1M, causing the precipitation of the product. After centrifugation, the solid was washed with water and dried under reduced pressure. Pure products were obtained in good yields.

Compound **HL_1_**. Yellow solid, yield = 90%. ^1^H-NMR (400 MHz, triethylammonium salt in D_2_O) δ (ppm) = 8.27 (s, 1H, H-4), 8.20 (s, 2H, Py), 7.70 (d, ^3^*J* = 7.57 Hz, ^4^*J* = 1.06 Hz, 1H, H-5), 7.61 (t, ^3^*J* = 7.75 Hz, ^4^*J* = 1.50 Hz, 1H, H-7), 7.38 (d, ^3^*J* = 7.49 Hz, 1H, H-8), 7.35(t, ^3^*J* = 7.57 Hz, 1H, H-6). ^13^C-NMR (150 MHz, triethylammonium salt in D_2_O) δ (ppm) = 177.6, 172.9, 165.7, 153.5, 148.7, 134.6, 130.7, 127.9, 125.3, 125.1, 121.2, 119.7, 114.5. HRMS [M − H]^−^
*m*/*z* = 310.03571, calculated for C_16_H_9_NO_6_
*m*/*z* = 311.04244. HPLC purity at 254 nm (triethylammonium salt) 98.5%, t_R_ = 5.69 min.

Compound **HL_2_**. Yellow solid, yield = 60%. ^1^H-NMR (400 MHz, *d*_6_-DMSO) δ (ppm) = 8.64 (s, 1H, H-4), 8.61 (s, 2H, Py), 8.25–8.24 (d, ^3^*J* = 2.03 Hz, 1H, H-5), 7.99–7.96 (dd, ^3^*J* = 2.27 Hz, 8.80 Hz, 1H, H-7), 7.33–7.31 (d, ^3^*J* = 8.84 Hz, 1H, H-8). ^13^C-NMR (150 MHz, *d*_6_-DMSO) δ (ppm) = 165.9, 159.1, 153.6, 149.1, 145.1, 142.6, 137.8, 126.7, 124.2, 121.8, 118.9, 88.9. HRMS [M − H]^−^
*m*/*z* = 435.93236, calculated for C_16_H_8_INO_6_
*m*/*z* = 436.94018. HPLC purity at 254 nm (triethylammonium salt) 98%, t_R_ = 6.54 min.

Complex **Na_3_[Eu(L_1_)_3_]**. Compound **HL_1_** (20 mg, 0.06 mmol, 1 eq) was dissolved in deionized water (30 mL) and NaOH 1M was added (130 mL, 2 eq), monitoring solution pH by litmus paper to be between 7 and 8. EuCl_3_·6H_2_O was added (5.5 mg, 0.02 mmol, 0.33 eq), the pH regulated to 7 and the mixture was stirred for 1h. The solution was then filtrated on a cellulose acetate filter (0.45 µm) and then concentrated to a volume of 1 mL and the precipitation of the complex was observed. After centrifugation, the solid was washed with water and dried under reduced pressure. The complex **Na_3_[Eu(L_1_)_3_]** was obtained as pure product in good yield (15 mg, yield = 61%). ^1^H-NMR (400 MHz, D_2_O) δ (ppm) = 7.41 (t, ^3^*J* = 7.85 Hz, 3H, H-7), 7.29 (d, ^3^*J* = 6.62 Hz, 3H, H-8), 7.19 (t, ^3^*J* = 7.11 Hz, 3H, H-6), 7.14 (d, ^3^*J* = 8.50 Hz, 3H, H-5), 6.96 (s, 3H, H-4), 4.33 (s, 6H, Py). HRMS [M − Na]^−^
*m*/*z* = 1125.98462, calculated for C_48_H_21_EuN_3_Na_2_O_18_^−^
*m*/*z* = 1125.98289. HPLC purity at 254 nm 96%, t_R_ = 5.80 min. Elemental analysis calculated (%) for C_48_H_21_EuN_3_Na_3_O_18_ × 0.15 NaCl × 0.5 H_2_O: C 50.19, H 1.84, N 3.66; found C 49.65, H 2.15, N 3.35.

Complex **Na_3_[Eu(L_2_)_3_]**. Compound **HL_2_** (20 mg, 0.05 mmol, 3 eq) was dissolved in deionized water (20 mL) and NaOH 1M was added (92 mL, 2 eq), monitoring solution pH by litmus paper to be between 7 and 8. EuCl_3_·6H_2_O was added (5.6 mg, 0.02 mmol, 1 eq), the pH regulated to 7 and the mixture was stirred for 1h. The solution was then filtrated on a cellulose acetate filter (0.45 µm) and then concentrated to a volume of 1 mL and the precipitation of the complex was observed. After centrifugation, the solid was washed with water and dried under reduced pressure. The complex **Na_3_[Eu(L_2_)_3_]** was obtained as pure product in good yield (12.5 mg, yield = 56%). ^1^H-NMR (400 MHz, *d*_6_-DMSO) δ (ppm) = 7.62 (s, 6H, Py), 7.60 (s, 3H, H-4), 6.93 (bs, 3H, H-8), 6.90 (bs, 3H, H-7), 6.89 (bs, 3H, H-5). HRMS [M − Na]^−^
*m*/*z* = 1503.67371, calculated for C_48_H_18_EuI_3_N_3_Na_2_O_18_^−^
*m*/*z* = 1503.67281. HPLC purity at 254 nm 95%, t_R_ = 6.70 min. Elemental analysis calculated (%) for C_48_H_18_EuI_3_N_3_Na_3_O_18_ × 0.45 NaCl × 0.3 H_2_O: C 37.77, H 1.19, N 2.75; found C 37.22, H 1.43, N 2.46.

### 3.2. NMR Titration of L_1_^2−^ with Eu^3+^ in D_2_O

A 1 mg sample of **HL_1_** solution in 0.5 mL of D_2_O (6.4 mM) with 2 eq of Et_3_N was titrated with increasing quantities of Eu^3+^ from a 40 mM solution of europium chloride hexahydrate in D_2_O. For each spectrum at least 100 scans were acquired, and the shimming re-done after each addition.

### 3.3. NMR Titration of L_1_^2−^ with Eu^3+^ in (CD_3_)_2_SO

A 1 mg sample of **HL_1_** solution in 1 mL of (CD_3_)_2_SO (3.2 mM) with 2 eq of Et_3_N was titrated with increasing quantities of Eu^3+^ from a 227 mM solution of europium chloride hexahydrate in D_2_O. For each spectrum at least 100 scans were acquired, and the shimming re-done after each addition.

### 3.4. Fluorescence/Lifetime Titration of L_1_^2−^ with Eu^3+^ in Tris HCl

Stock solution 1 mg of **HL_1_** in 10 mL of Tris HCl 10 mM (320 mM), diluted to 1.1 µM with Tris HCl 10 mM. The latter titrated (in 1 cm cell) with a 3.2 mM solution of europium chloride hexahydrate in water (Eu^3+^ concentration determined by ICP-MS).

### 3.5. Fluorescence Spectra in DMSO of Eu^3+^/L_1/2_^2−^ Mixture of 1:2, 1:3 and 1:4 Molar Ratios

Stock solution of 1 mg of **HL_1_** (0.32 mM) or **HL_2_** (0.23 mM) in 10 mL of DMSO with 2 eq of Et_3_N; on a 1 mL amount of stock were respectively added 0.25, 0.33 and 0.5 eq of Eu^3+^ (chloride hexahydrate) from a 8.3 mM stock in water (Eu^3+^ concentration determined by ICP-MS).

### 3.6. Absorption and Emission Spectra of L_1_^2−^, L_2_^2−^, Na_3_[Eu(L_1_)_3_] and Na_3_[Eu(L_2_)_3_]

Stock solution for **Na_3_[Eu(L_1_)_3_]** 1 mg in 10 mL of water (0.087 mM) diluted 1:100 in water for both absorption and fluorescence spectra acquisition (1 cm cell). Stock solution for **Na_3_[Eu(L_2_)_3_]** 1 mg in 10 mL of water (0.065 mM) diluted 1:100 in water for both absorption and fluorescence spectra acquisition (1 cm cell). Stock solution of 1 mg of **HL_1_** (0.32 mM) or **HL_2_** (0.23 mM) in 10 mL of DMSO with 2 eq of Et_3_N diluted both 1:300 with water for the absorption spectra (1 cm cell). Both lifetime and excitation spectra recorded with the aforementioned concentration and monitoring on the 615 europium emission band.

### 3.7. Quantum Yield Determination of Na_3_[Eu(L_1_)_3_] and Na_3_[Eu(L_2_)_3_] Complexes

For the determination of the quantum yield of the complex (Φ^L^_Eu_), we measured the absorbance and fluorescence of several solutions of quinine sulfate in 0.5 M H_2_SO_4_ (Φ_QS_ = 0.56) and of the tris complexes in water. Note that solution absorbances were in all cases below 0.1. Stock solution for **Na_3_[Eu(L_1_)_3_]** 0.88 mg in 5 mL DMSO (0.15 mM) diluted in DMSO or water. Stock solution for **Na_3_[Eu(L_2_)_3_]** 1.2 mg in 5 mL DMSO (0.15 mM) diluted in DMSO or water.

### 3.8. Sensitization Parameter (η_sens_) for Na_3_[Eu(L_1_)_3_] and Na_3_[Eu(L_2_)_3_] Complexes

According to theory [47], the radiative lifetime *τ_r_* of Eu^3+^ can be calculated from the integrated total emission I_tot_ of the ^5^D_0_ → ^7^F_J_ transitions (J = 0–6) and the integrated emission I_MD_ of magnetic dipole transition ^5^D_0_ → ^7^F_1_, according to Equation (3):(3)$$k_{r}=\frac{1}{\tau_{r}}\;=\;A_{MD,0}\cdot n^{3}\cdot \left(\frac{I_{tot}}{I_{MD}}\right)$$
where *A_MD_*_,0_ = 14.65 s^−1^ is the spontaneous emission probability of the ^5^D_0_→^7^F_1_ transition and n the refractive index of the medium. We found *τ_r_* = 4.1 ms for **Na_3_[Eu(L_1_)_3_]** in water and *τ_r_* = 5.2 ms for **Na_3_[Eu(L_2_)_3_]** in water. The intrinsic quantum yield of the lanthanide ion $\Phi_{\mathrm{Eu}}^{\mathrm{Eu}}$ (i.e., the quantum yield upon direct 4f−4f excitation) is related to the luminescence and radiative lifetimes according to Equation (4):(4)$$\Phi_{\mathrm{Eu}}^{\mathrm{Eu}}=\tau /\tau_{r}$$

This leads to $\Phi_{\mathrm{Eu}}^{\mathrm{Eu}}$ = 0.15 for Na_3_[Eu(L_1_)_3_] and $\Phi_{\mathrm{Eu}}^{\mathrm{Eu}}$ = 0.11 for Na_3_[Eu(L_2_)_3_]. The photoluminescence yield of the complex upon ligand excitation ($\Phi_{\mathrm{Eu}}^{L}$) is simply the product of $\Phi_{\mathrm{Eu}}^{\mathrm{Eu}}$ and the sensitization efficiency *η_sens_* of the antenna system Equation (5):(5)$$\Phi_{\mathrm{Eu}}^{L}=\eta_{sens}\cdot \Phi_{\mathrm{Eu}}^{\mathrm{Eu}}$$

Then, for **Na_3_[Eu(L_1_)_3_]** in water *η_sens_* = 0.0032 and for **Na_3_[Eu(L_2_)_3_]**
*η_sens_* = 0.0039.

## 4. Conclusions

In conclusion, we synthesized a new series of self-assembled tris europium complexes, with ligands designed ad hoc to merge the versatility of the coumarin scaffold as antenna and the chemical stability of 2,6-dipicolinate as ligand. The fully conjugated coumarin-dipicolinate ligands **L_1_** and **L_2_** succeeded in the task of sensitizing the europium ion: with **Na_3_[Eu(L_1_)_3_]** and **Na_3_[Eu(L_2_)_3_]**, which displayed low quantum yields and good lifetimes in water, we successfully realized a total control of the ligand/metal-centered emission with a simple chemical modification (the insertion of an iodine atom on the coumarin core). In addition, we encountered a rich and dynamic chemical speciation rationalized by paramagnetic NMR and emission spectroscopy studies: this synergistic approach allowed us to depict a detailed picture of the solution chemistry of this new system, and to found a first example of 1:4 stoichiometry dpa-based complex, which is currently under investigation. Moreover, despite the low photoluminescence quantum yield, the simple and versatile synthesis of the coumarin-dpa ligand opens the possibility of a future screening of different coumarin antennas for the fine optimization of the emission properties, with the final aim of anchoring the ligand fragment on macrocycle scaffolds to realize sensitive intracellular molecular probes. Additionally, the usual absorption range of coumarin derivatives in the near UV region of the electromagnetic spectrum opens the way to the development of more biologically compatible lanthanide-based probes.

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
