# Peer review of "New Coumarin Dipicolinate Europium Complexes with a Rich Chemical Speciation and Tunable Luminescence"

_molecules, 2021, doi:10.3390/molecules26051265_

Round 1
Reviewer 1 Report
Di Pietro et al reported on the synthesis and optical properties of two trivalent Eu complexes whose coumarin ligands vary by a H/I substituent. The authors are motivated by the lack of coumarin-based lanthanides complexes owing to poor photo-sanitization of the metal center. They find that substitution of H for I leads to a dramatic loss of ligand based fluorescence (presumably a πàπ* transition). The findings represent another example of the heavy atom effect on sensitization of trivalent lanthanide ions. While I can only speak to the photophysics, the article is overall well written and the results are clearly presented. The editor, however, should ensure that a reviewer with structural characterization experience via NMR has approved this manuscript. Minor spelling and grammatical errors are present (i.e line 38 ‘phtotophysical’ and line 611 ‘sensitize’ is not grammatical correct), but overall the manuscript is sound. As such, I recommend acceptance of the manuscript to MDPI Molecules.
Author Response
Reviewer 1
Di Pietro et al reported on the synthesis and optical properties of two trivalent Eu complexes whose coumarin ligands vary by a H/I substituent. The authors are motivated by the lack of coumarin-based lanthanides complexes owing to poor photo-sanitization of the metal center. They find that substitution of H for I leads to a dramatic loss of ligand based fluorescence (presumably a π-π* transition). The findings represent another example of the heavy atom effect on sensitization of trivalent lanthanide ions. While I can only speak to the photophysics, the article is overall well written and the results are clearly presented. The editor, however, should ensure that a reviewer with structural characterization experience via NMR has approved this manuscript. Minor spelling and grammatical errors are present (i.e line 38 ‘phtotophysical’ and line 611 ‘sensitize’ is not grammatical correct), but overall the manuscript is sound. As such, I recommend acceptance of the manuscript to MDPI Molecules.
We thank the reviewer, we have fixed the minor spelling as requested.
Reviewer 2 Report
The paper reports the synthesis of coumarine-based europium (III) complexes. The introduction and the purpose of this work are clear. The synthesis and NMR characterizations are well conducted. I have more reserve on the photoluminescence characterization, which are more difficult to understand due to a lack of clarity of the presentation (information missing).
(1) the in situ fluorescence measurement upon titration is presented on Figure S24. In the manuscript, the authors state “The variation in intensity of the 615 nm band upon the addition of europium, clearly indicates the formation of the 1:3 complex [Eu(L1)3]3”. The authors monitor the fluorescence of Eu3+ which increases with the formation of the 1:3 complex (up to 0.3 eq Eu3+). It is not clear why the fluorescence intensity decreases after this value and why there is a plateau around 0.5-0.6 eq Eu3+
(2) the authors report the fluorescence decay of the complexes (Figure S25). The fluorescence decay curves should be plotted on a logarithmic scale for the y-coordinate so that the mono-exponential character of the decay could be evaluated. All figures from S25 to S37 should be modified.
(3) The legend of Figure S26 is wrong and should be corrected.
(4) for sake of clarity, the authors should add a figure with the photoluminescence spectra of the different identified species: tetrakis, tris, bis. How different is the fluorescence of Eu3+ in these species? I would not expect any difference in the shape of Eu3+ emission bands. How do the authors manage to separate the contribution of the different complexes (terakis, bis, tris) ? And how do they plot Figure 3?
(5) What is the difference between Figure 4 and Figure S38 ? This is not clear to me.
(6) Figure S42 and S43 should be plotted with a logarithmic scale for the y-axis.
(7) the discussion on the quantum yield is unclear: in this work the authors find a efficiency of transfer from the ligand to Eu3+ of 0.3%, which is 10 times less than other molecules reported in the literature (2%). Can they give an explanation? The internal quantum yields are low : 0.04%. Can the authors comment these values ? Are these complexes of any use with these low quantum yield values ?
(8) in the conclusion, the authors say that their systems show “modest” quantum yield (0.04%). To me, the word “modest” is not adapted.
Author Response
The paper reports the synthesis of coumarine-based europium (III) complexes. The introduction and the purpose of this work are clear. The synthesis and NMR characterizations are well conducted. I have more reserve on the photoluminescence characterization, which are more difficult to understand due to a lack of clarity of the presentation (information missing).
(1) the in situ fluorescence measurement upon titration is presented on Figure S24. In the manuscript, the authors state “The variation in intensity of the 615 nm band upon the addition of europium, clearly indicates the formation of the 1:3 complex [Eu(L1)3]3”. The authors monitor the fluorescence of Eu3+ which increases with the formation of the 1:3 complex (up to 0.3 eq Eu3+). It is not clear why the fluorescence intensity decreases after this value and why there is a plateau around 0.5-0.6 eq Eu3+
We thank the reviewer for this observation; the fluorescence profile beyond 0.3 eq is characterized by an initial coexistence of tris and bis complexes, in particular between 0.35 eq and 0.6 eq (NMR in Figure 2 and lifetime in Figure 3). After 0.6-0.65 eq we envisaged the sole bis species. The 1:2 complex has a lower propensity to luminesce with respect to tris (and tetrakis) because it has more water molecules coordinated, then the constant decrease of luminescence at 615 nm. We believe that the 0.45-0.6 eq interval in Figure S24 is part of a constant decrease of emission, and the 0.6 eq point fell within the experimental error of the measurement. We added a sentence along this line in the main text.
(2) the authors report the fluorescence decay of the complexes (Figure S25). The fluorescence decay curves should be plotted on a logarithmic scale for the y-coordinate so that the mono-exponential character of the decay could be evaluated. All figures from S25 to S37 should be modified.
We added all the logarithmic y-axis plots for each lifetime measurement.
(3) The legend of Figure S26 is wrong and should be corrected.
We checked the legend in Figure S26, but we did not envisaged any error.
(4) for sake of clarity, the authors should add a figure with the photoluminescence spectra of the different identified species: tetrakis, tris, bis. How different is the fluorescence of Eu3+ in these species? I would not expect any difference in the shape of Eu3+ emission bands. How do the authors manage to separate the contribution of the different complexes (terakis, bis, tris) ? And how do they plot Figure 3?
We thank the reviewer for raising this question: Figure 4 for ligand L1 (and Figure S38 for ligand L2) shows the luminescence profile for 1:4, 1:3 and 1:2 molar ratio mixtures of Europium and ligand (triethylammonium salt). At these molar ratios there is the predominance of tetrakis, tris and bis complex respectively. The reviewer is correct about the quasi-identical profile of the europium bands for tetrakis and tris (their symmetry is characterized by similar multiplicity for each band), but especially for L1 at 1:2 ratio, the bis complex (C2 symmetry) showed different multiplicity in accordance to its lower symmetry. Lastly, Figure 3 is obtained plotting the normalized I0 values (see Eq 2) taken from each lifetime fitting for the L1 titration: this analysis allowed us to rationalize the isolated contribution of the single species, especially in intervals where there is a coexistence of them, thanks to a bi-exponential fitting. Interestingly, this speciation picture is in accordance with what we found in the NMR titration.
(5) What is the difference between Figure 4 and Figure S38 ? This is not clear to me.
Figure 4 is referred to ligand L1, while Figure S38 is referred to L2.
(6) Figure S42 and S43 should be plotted with a logarithmic scale for the y-axis.
We added the new plots according to the reviewer suggestion.
(7) the discussion on the quantum yield is unclear: in this work the authors find a efficiency of transfer from the ligand to Eu3+ of 0.3%, which is 10 times less than other molecules reported in the literature (2%). Can they give an explanation? The internal quantum yields are low : 0.04%. Can the authors comment these values ? Are these complexes of any use with these low quantum yield values ?
We thank the reviewer for these comments. This first work on this new system has been focused on assess if these ligands are capable on sensitize europium ions and, considering previous literature attempts, if we could control with a simple chemical modification (from L1 to L2) the dual vs. mono emission for coumarin antenna based europium complexes. While from this side we achieved our purpose, we however encountered a low sensitization efficiency and a low europium photoluminescence quantum yield for both complexes. As we underlined in the paper, there are two strong reasons for this result: the first is related to the known low propensity for a coumarin scaffold to perform an efficient intersystem crossing process (L1 is the perfect example), second is the energy matching between the ligand centered triplet energy state and the Eu3+ 5D0–7F0 excited one. For both ligands we believed they lie outside the 2500–3500 cm−1 optimum range for a performant energy transfer. It is important to underline that we specifically designed this new system in such a way which will allow us to easily synthesize different coumarin cores, to finely tune the energy matching and then to improve the quantum yield. This work represents a fundamental step towards that direction.
(8) in the conclusion, the authors say that their systems show “modest” quantum yield (0.04%). To me, the word “modest” is not adapted.
In accordance with the reviewer suggestion we changed the word “modest” with “low”.
Reviewer 3 Report
This manuscript reports on the study of the luminescence of europium complexes with two derivatives of helidamic acid and coumarin as ligands. The сompositions of the complexes formed in solutions with various concentrations of ligands as well as that of the solid substances were determined by means of NMR and MS methods. Due to the conjugation between the coumarin and pyridinedicarboxylic moieties, the coumarin fragment serves as an antenna exciting the luminescence of the lanthanide ion. The presence of an iodine substitute in the coumarin fragment affects the nature of the luminescence, stabilizing the excited triplet state. The paper can be published after minor corretions.
Abstract. "We also encountered a rich solution chemical speciation, studied in details by means of paramagnetic NMR and emission spectroscopy" - this sentence should be rephrased. "Rich chemical speciation in solution" would be better.
Line 67. "The synthetic pathway for the preparation of the new ligands HL1 and HL2 and the relative tris europium complexes Na3[Eu(L1)3] and Na3[Eu(L2)3] is leveraged on the key intermediate triester 3 that makes possible to conjugate virtually any coumarin scaffold to the dpa ring with simple and inexpensive chemistry." "The synthetic pathway ... is leveraged on the key intermediate triester" - unfortunately, I can't understand what it means.
Line 72: "the aromatic/heteroaromatic portion of the coumarin side of the ligand possesses a salicyl aldehyde-like functionality." Coumarin moiety is formed from salicyl aldehyde, but it does not have a salicyl aldehyde-like functionality. This should be corrected.
Line 202. The authors state that "the solid precipitating from water at similar stoichiometry ratio can be then referred to the 1:2 complex, which is less soluble compared with the tris compound, considering that it is a mono-charged system." Why haven't the authors charaterized those solids by elemental analysis or, for example, by thermogravimetry in air (which would lead to Na2CO3 and Eu2O3 mixture, and the mass loss could be compared to theoretical one) to prove their composition? The absence of elemental analyses in the synthetic part is also strange.
The part "Chemical Speciation of the Eu/L1/2 System" should be divided in two chapters, one for NMR, and another for luminescence study.
The conclusions are too humble, they should be developed to reflect adequately the rich findings of luminescent studies.
Author Response
This manuscript reports on the study of the luminescence of europium complexes with two derivatives of chelidamic acid and coumarin as ligands. The composition of the complexes formed in solutions with various concentrations of ligands as well as that of the solid substances were determined by means of NMR and MS methods. Due to the conjugation between the coumarin and pyridinedicarboxylic moieties, the coumarin fragment serves as an antenna exciting the luminescence of the lanthanide ion. The presence of an iodine substitute in the coumarin fragment affects the nature of the luminescence, stabilizing the excited triplet state. The paper can be published after minor corrections.
Abstract. "We also encountered a rich solution chemical speciation, studied in details by means of paramagnetic NMR and emission spectroscopy" - this sentence should be rephrased. "Rich chemical speciation in solution" would be better.
We changed the text in accordance to the reviewer suggestion.
Line 67. "The synthetic pathway for the preparation of the new ligands HL1 and HL2 and the relative tris europium complexes Na3[Eu(L1)3] and Na3[Eu(L2)3] is leveraged on the key intermediate triester 3 that makes possible to conjugate virtually any coumarin scaffold to the dpa ring with simple and inexpensive chemistry." "The synthetic pathway ... is leveraged on the key intermediate triester" - unfortunately, I can't understand what it means.
We thank the reviewer for the observation, we changed the term “leveraged” with “designed around” which is more adapt for the comprehension of the sentence.
Line 72: "the aromatic/heteroaromatic portion of the coumarin side of the ligand possesses a salicyl aldehyde-like functionality." Coumarin moiety is formed from salicyl aldehyde, but it does not have a salicyl aldehyde-like functionality. This should be corrected.
We thank the reviewer for this comment, because it was not clear the meaning of the sentence. We changed the phrase to “This planning enables a highly versatile synthesis, because the only requirement being that the aromatic/heteroaromatic portion needed for the coumarin construction possesses a salicyl aldehyde-like functionality.”.
Line 202. The authors state that "the solid precipitating from water at similar stoichiometry ratio can be then referred to the 1:2 complex, which is less soluble compared with the tris compound, considering that it is a mono-charged system." Why haven't the authors characterized those solids by elemental analysis or, for example, by thermogravimetry in air (which would lead to Na2CO3 and Eu2O3 mixture, and the mass loss could be compared to theoretical one) to prove their composition? The absence of elemental analyses in the synthetic part is also strange.
We thanks the reviewer for these questions: the aim of this work is the synthesis, isolation and characterization of the tris complexes. The other species encountered in the chemical speciation study will be object of future investigations, in particular for the tetrakis species. Importantly, we added the elemental analysis for the two target complexes as requested in the experimental part, which strongly improve the purity assessment.
The part "Chemical Speciation of the Eu/L1/2 System" should be divided in two chapters, one for NMR, and another for luminescence study.
We modified the text in accordance with the reviewer suggestion.
The conclusions are too humble, they should be developed to reflect adequately the rich findings of luminescent studies.
We thanks the reviewer for this comment; we enriched the conclusion to better reflect the findings of the work.
Round 2
Reviewer 2 Report
Thanks for the reply.